# Cost-Effectiveness of the Comprehensive Interdisciplinary Program-Care in Informal Caregivers of People with Alzheimer’s Disease

**DOI:** 10.3390/ijerph192215243

**Published:** 2022-11-18

**Authors:** Laura Muñoz-Bermejo, María José González-Becerra, Sabina Barrios-Fernández, Salvador Postigo-Mota, María del Rocío Jerez-Barroso, Juan Agustín Franco Martínez, Belén Suárez-Lantarón, Diego Muñoz Marín, Nieves Martín-Bermúdez, Raquel Ortés-Gómez, Martín Gómez-Ullate-García de León, Marta Martínez-Acevedo, Lara Rocha-Gómez, Sara Espejo-Antúnez, Mercedes Fraile-Bravo, María Gloria Solís Galán, Ignacio Chato-Gonzalo, Francisco Javier Domínguez Muñoz, Miguel Ángel Hernández-Mocholí, Miguel Madruga-Vicente, Angelina Prado-Solano, María Mendoza-Muñoz, Jorge Carlos-Vivas, Jorge Pérez-Gómez, Raquel Pastor-Cisneros, Paulina Fuentes-Flores, Damián Pereira-Payo, Javier De Los Ríos-Calonge, Javier Urbano-Mairena, Joan Guerra-Bustamante, José Carmelo Adsuar

**Affiliations:** 1Social Impact and Innovation in Health (InHEALTH) Research Group, University Centre of Mérida, University of Extremadura, 06800 Mérida, Spain; 2BioẼrgon Research Group, University of Extremadura, 10003 Cáceres, Spain; 3Department of Nursing, Faculty of Medicine, University of Extremadura, 06006 Badajoz, Spain; 4Health Economy Motricity and Education (HEME) Research Group, Faculty of Sport Science, University of Extremadura, 10003 Cáceres, Spain; 5Education Sciences Department, Faculty of Education, University of Extremadura, 06006 Badajoz, Spain; 6Department of Musical, Plastic and Corporal Expression, Faculty of Sport Sciences, University of Extremadura, Av. de la Universidad, s/n, 10003 Cáceres, Spain; 7Department of Educational Sciences, Faculty of Education and Education and Psychology, University of Extremadura, 10003 Cáceres, Spain; 8Area Specialist in the Extremadura Health Service, Geriatrics Service of the Hospital Virgen del Puerto de Plasencia, 10600 Plasencia, Spain; 9Department of Teaching of Musical, Plastic and Body Expression, Faculty of Teacher Training, University of Extremadura, 10004 Cáceres, Spain; 10Neurology Service, Badajoz University Hospital Complex, 06080 Badajoz, Spain; 11Gpex-Eshaex Superior School of Hotel Management and Agrotourism of Extremadura, 06800 Mérida, Spain; 12Department of Educational Sciences, Faculty of Teacher Training, University of Extremadura, 10004 Cáceres, Spain; 13Department of Social Sciences, Language and Literature Teaching, Faculty of Teacher Training, University of Extremadura, 10004 Cáceres, Spain; 14Physical Activity and Quality of Life (AFYCAV) Research Group, Faculty of Sport Science, University of Extremadura, 10003 Cáceres, Spain; 15Physical and Health Literacy and Health-Related Quality of Life (PHYQOL) Research Group, Faculty of Sport Sciences, University of Extremadura, 10003 Cáceres, Spain; 16Promoting a Healthy Society (PHeSo) Research Group, Faculty of Sport Sciences, University of Extremadura, 10003 Caceres, Spain

**Keywords:** Alzheimer’s caregivers, training programs, quality of life

## Abstract

People with Alzheimer’s disease (AD) diagnosis who get informal care remain at home longer, reducing the demand for healthcare resources but increasing the stress of caregiving. Research on the effectiveness of physical training, psychoeducational, cognitive–behavioural, and health education programs in reducing the caregiver load and enhancing health-related quality of life (HRQoL) exist, but none exist about an integrated interdisciplinary program. The goals of this project are (1) to assess the Integral-CARE Interdisciplinary Program (IP) applicability, safety, effects on HRQoL, and the incremental cost-effectiveness ratio for AD caregivers; (2) to evaluate the IP applicability and cost-effectiveness to enhance the physical, psychoemotional, cognitive–behavioural dimensions, and the health education status of informal caregivers, and (3) to study the transference of the results to the public and private sectors. A randomized controlled trial will be conducted with an experimental (IP) and a control group (no intervention). The PI will be conducted over nine months using face-to-face sessions (twice a week) and virtual sessions on an online platform (once a week). There will be an initial, interim (every three months), and final assessment. Focus groups with social and health agents will be organized to determine the most important information to convey to the public and private sectors in Extremadura (Spain). Applicability, safety, HRQoL, incremental cost-effectiveness ratio, and HRQoL will be the main outcome measures, while secondary measures will include sociodemographic data; physical, psychoemotional, health education, and cognitive–behavioural domains; program adherence; and patient health status. Data will be examined per procedure and intention to treat. A cost-effectiveness study will also be performed from the viewpoints of private and public healthcare resources.

## 1. Introduction

The quality of life of the world’s population has led to an increase in life expectancy. Advanced age increases the risk of dementia, with a consequent increase in elderly care demand, particularly in western developed countries [1]. Thus, it is estimated that there are about 55 million people with dementia in the world and, according to population projections, this number is expected to double every 20 years [2].

Alzheimer’s disease (AD) is a severe neurodegenerative condition and the most frequent cause of Dementia in the elderly [3], accounting for 60–80% of all cases [4] and representing a leading cause of mortality and disability worldwide [5]. In the early preclinical and mild cognitive impairment stages, this condition has little impact on daily life activities, but its progression to AD eventually results in cognitive and memory impairment and profound loss of independence and functionality [6]. Thus, the need for support and care increases as the disease progresses and this is mainly provided by family members, mainly wives and daughters [7,8]. These informal primary caregivers face caregiving tasks that can last a long time [9]. Moreover, AD caregivers usually provide care for more years than others [10]. Hence, caring for a person with AD requires compelling dedication and commitment due to the irreversible and progressive nature of the disease, its long duration and the deterioration in cognitive, behavioural, and functional skills [8]. Caregivers are also likely to experience an increase in out-of-pocket expenses, which reduces their ability to save [11]. Thus, these family caregivers report high levels of burden, as well as health issues and, consequently, perceive a greater need for support and assistance [9,12,13]. Despite these perceived needs, they often reject recommended support services and only make use of them when they are no longer able to cope psychologically or emotionally with the care situation [14]. Social support, perception or appraisal of stressors, and coping strategies used [15] are also factors that have traditionally been related to caregiver stress and, consequently, to the erosion of resilience [16].

In terms of therapeutic intervention, it has been shown that the improvement of psycho-emotional symptoms in the caregiver, up to 50% in the case of spouses, can delay the patient’s admission, which should stimulate the implementation of multidimensional programs [14]. In this sense, and to prevent stress risk factors, early identification of cognitive impairment through protocolized disease screening, together with health education activities, would allow caregivers and their families to receive care at an earlier stage of the disease process, which could facilitate health, financial, and legal decision-making while the patient still retains the capacity to do so [17]. In terms of stressor prevention, it has been shown in different studies that caregivers often do not have time to participate in preventive health activities, such as regular physical training [18,19]. However, we know that physical exercise has positive effects on caregiver burden [20,21]. In caregivers of people with dementia, physical activity can also have direct psychological benefits and can reduce subjective burden [22].

It is important to consider that the dependency of AD individuals varies throughout their evolution and their caregiver’s dedication. Consequently, direct medical costs and caregiver burden increase with the severity of cognitive impairment from the onset to the advanced stages of AD [23]. For public health and social services, the economic cost increases exponentially as cognitive deterioration increases [24], as does the use of social and healthcare resources, although most of the cost of AD is borne by the family [25,26,27]. Socio-healthcare for people with AD and other dementias in Spain has an average cost of 24,184 euros per patient per year, but up to 71% of this cost (about 15,724 euros per patient per year) is borne by the family [28]. However, economic analyses to measure the cost-effectiveness of AD care overlook many costs, such as medical care for caregivers, reduced HRQoL, and hidden costs that accrue before diagnosis [29]. The correct estimated cost of AD will have an enormous impact on our current fragile support system and should also be reflected in the implementation of future preventive strategies [11].

Specifically, one study predicted a DA-related tax loss of €74,288, most of which was incurred within ten years of the start of the DA. Most of the costs were due to direct employment-related tax losses (€35,925). This was mainly due to caregivers who had to reduce or give up work to provide informal care. Over 10.5 years, caregivers’ incomes were predicted to be reduced by €56,967 compared to their non-caring counterparts with AD [30]. The rising cost of dementia care and the scarcity of resources has promoted the need to assess the economic feasibility of any intervention. Evaluation of health interventions and cost-effectiveness assessments represent one component of a strategy to make healthcare and interventions more accessible to those who need them. Progress in health and social care depends on the efficient use of resources, including knowledge about what interventions and strategies work, how much they cost and how they are managed and implemented [31,32]. In this sense, healthcare decisions to improve individual and collective health status must consider not only clinical effectiveness but also economic effectiveness.

Some studies intervene through psycho-educational [33,34] or cognitive–behavioural and educational [35,36] programs in caregivers of persons with AD or publications on didactic exercise interventions in persons with AD and their caregivers [37,38]. However, no study has evaluated the incremental cost-effectiveness ratio of an IP with psychoeducational, cognitive–behavioural, educational, and physical training interventions in primary caregivers of persons with AD. This project would be a pioneer in assessing the costs and effectiveness of a comprehensive program.

The implications for the health, quality of life and economy of the patient, his or her family, and the health system demand the implementation of the latest, most effective, and least costly comprehensive interventions. Therefore, the primary goals of this project are (1) to examine the applicability, safety, and incremental cost-effectiveness ratio of PI Integral Care for caregivers of people with AD; (2) to examine the most practical and cost-effective PI Integral-CARE alternative to enhance the health-related quality of life in informal caregivers of people with AD; and (3) to determine the crucial elements for its transfer to the public and private social-health system, assessing the acceptance of PI Integral Care for AD caregivers in the social-health system’s representatives in charge of integrating it in their service offer. The secondary goals of this experiment are to assess how PI Integral-CARE affects the physical, psychological, cognitive–behavioural, and educational areas of the participants.

## 2. Materials and Methods

### 2.1. Study Design

The methodology followed is the Consolidated Standards of Reporting Trials Statement (CONSORT) for randomized controlled trials [39], as well as the recommendations for the conduct, methodological practices, and reporting of cost-effectiveness analyses of the second panel on cost-effectiveness in health and medicine [40].

A parallel-group randomized controlled trial will be conducted comprising a 9-month intervention phase, followed by a 1-month follow-up or observation phase with controls every 3 months, including an evaluation at the beginning and another at the end of the program. Caregiver participants will be randomly assigned to the intervention group (experimental) or the “usual care” group (control). The measurements will be carried out in the centres of the associations or social-healthcare centres. The elderly caregiver recruitment will be carried out by the social workers in their communities.

### 2.2. Ethics Approval

Ethical approval was granted by the Bioethics and Biosafety Committee of the University of Extremadura (approval number: 129/2020). This study has been registered in The Clinical Trials Register provided by the Australian and New Zealand Clinical Trials Registry (Application number: 378330; https://www.anzctr.org.au/, accessed on 15 September 2022).

### 2.3. Sample Size

The number of participants to be included in the study was calculated based on the change in the quality of life assessed with the EQ-5D-5L questionnaire. To the best of our knowledge, no data are available on the minimum actual change in the EQ-5D-5L in informal caregivers, so a minimum actual difference of 0.07 [41] was used as a reference. Therefore, a total of 50 participants (25 in the experimental group and 25 in the control group) are needed, accepting an alpha risk of 0.05 and a beta risk of 0.2 in a bilateral contrast, 25 subjects in the first group and 25.0 in the second group are needed to detect a difference equal to or greater than 0.07 units. The common standard deviation is assumed to be 0.12 [42], with a correlation coefficient between the initial and final measurement of 0.8. A loss-to-follow-up rate of 25% was estimated. Version 7.12 of the GRANMO sample size calculator (https://www.imim.es/ofertadeserveis/software-public/granmo/, accessed on 17 February 2022) was used.

### 2.4. Randomization and Blinding

Once the initial assessments are completed, all participants will be randomly assigned to either the experimental or control intervention groups. Before enrolling participants, Research Randomizer software (version 4.0, Geoffrey C. Urbaniak and Scott Plous, Middletown, CT, USA; http://www.randomizer.org, accessed on 16 June 2022) [43] will be used to create a randomisation sequence to assign participants to either the experimental or control group (1:1). The randomisation sequence will be prepared by a member of the research team with no clinical involvement in the trial. The allocation will be hidden in a password-protected computer file. Participants will be aware of their group assignment, while outcome assessors and data analysts will be blinded to the assignment.

#### Individual Participant Data Sharing (IPD) Statement

Data sets generated and/or analysed during the present study are/will be made available upon request to the principal investigator. Data will be available for 2 years. All data will be recorded when measurements are taken. The data for the thematic measures mentioned are either continuous or categorical variables. The data will be coded to be fully anonymised so that all research groups can work with the data for analysis. Data entry will be performed in duplicate. Only the principal investigator will know the code and will keep the originals safe.

### 2.5. Participants

Informal caregivers of people with AD will be recruited from specific associations and social health centres using non-probability convenience sampling. Potential participants will be identified by the research team and screened for eligibility based on the inclusion criteria below.

i. Performing the role of informal primary caregiver of a person with AD. Care for the ill person for more than 20 h per week for more than 3 months and intend to continue for the next 12 months.

ii. Not having pathology that contraindicates the exercise program or requires special attention (coronary pathologies, thrombosis, bone, kidney, moderate or severe pulmonary pathologies, symptoms associated with COVID-19, etc.). The questionnaire for the practice of physical activity and sport PAR-Q [44] will be administered to check if the patient suffers from diseases that impede the physical load.

iii. Not having carried out any physical exercise program in the 3 months before the intervention.

iv. Not to have participated/received psychoeducational and cognitive–behavioural sessions in the 3 months before the intervention.

v. Provide a signed informed consent form for the study.

### 2.6. Interventions

(a) Experimental Group (Integral-CARE): designed to combine face-to-face and virtual interventions. For 9 months, two weekly face-to-face sessions and one weekly virtual session will be developed employing “training pills” through the online platform designed for the program. The sessions will address three fundamental areas in the field of care: physical, psychoemotional and cognitive–behavioural areas, and health education. Before the initial assessment of each participant, an open, semi-structured and iterative interview will be conducted to minimize the bias of misunderstanding the questions in the questionnaires.

Face-to-face interventions:

Each face-to-face physical training session will last approximately 60 min, beginning with 10 min of warm-up activities followed by 40 min of core practice and, finally, 10 min of cool-down exercises. When face-to-face sessions occupy other areas, they will be 50 min in duration. All physical testing will be conducted under the dual-task paradigm. This is a procedure that requires an individual to perform two tasks simultaneously, to compare performance with single-task conditions. Motor tasks are combined with cognitive tasks, which involve a rapprochement between everyday life and the clinical setting [45]. The dual-task interference paradigm shows that the performance of two simultaneous tasks results in competition between the available attentional resources, leading to a decrease in performance on both tasks [46]. Dual tests allow revealing the existence of cognitive or motor alterations, even if subtle, that may be found to be related to the onset or worsening of diseases [47].

Virtual sessions:

Virtual sessions will be provided to participating caregivers through access to the online platform created for Integral-CARE. They will have a duration of 15 min and will be asynchronous so that participants can access the training at any time. The planning of the interventions is presented in Table 1:

Safety of the experimental group intervention

Furthermore, to guarantee the participants’ safety, all sessions will be supervised and/or directed by professionals in Physical Activity and Sport Sciences, Education, Nursing, Anthropology, Psychopedagogy, Geriatrics or Occupational Therapy. Evaluation and intervention sessions will take place in the associations’ day centres or socio-health centres, equipped with approved material, and attended by the professionals of the facilities. In addition, during the sessions, there will always be two cell phones to attend to professionals or participants in case of any incident.

Health Emergency Situation

If the case of COVID-19 spread to other health emergencies, the Integral-CARE program is prepared to adapt to a virtual modality. Initial planning sessions will be available on the Integral-CARE virtual platform. For this purpose, professionals in each intervention area will conduct videos of the planned sessions (15 min).

(b) Control Group: will continue performing their usual tasks (“usual care”).

### 2.7. Measures and Procedures

A variety of instruments will be used to assess the feasibility and effectiveness of this comprehensive program (Table 2). The assessments will be conducted at baseline, at 3, 6 and 9 months, and 1 month after the intervention.

Sociodemographic data and anthropometric measurements will be obtained to characterize the sample under study using various tests, measures, and questionnaires to evaluate the effect of the intervention from the different areas addressed. The questionnaires will be self-administered, although to include people with reading and writing problems, sensory problems, etc., the evaluators will provide help to the person who requires it to solve any doubts or questions that may arise. The questionnaires from the psychoemotional and cognitive–behavioural areas and those related to health education will be studied to evaluate the impact of the project on men and women.

#### 2.7.1. Randomized Trial Main Measures

Applicability. This will be calculated as the percentage of participants who can perform the proposed activities and training. If any participant is unable to perform the intervention, the cause will be noted.

Safety. A record will be kept for each of the sessions in which any incident, injury or problem that arises will be noted, recording the possible origin of the problem.

HRQoL will be assessed through the following questionnaires:EQ-5D-5L [48]. A questionnaire that assesses health status, first in levels of severity by dimensions (descriptive system) and subsequently through a visual analogue scale (VAS). The third element of the questionnaire is the index of social values obtained for each health state generated by the instrument. The descriptive system contains five health dimensions (mobility, self-care, activities of daily living, pain/discomfort, and anxiety/depression) and each dimension has five levels of severity (no problems, mild problems, moderate problems, severe problems, and extreme problems/impossibility). The algorithm for calculating the EQ-5D-5L utility index will be the ‘’cross-walking’’ of the Spanish version of the EuroQol of levels. Each participant indicates the level that best reflects his or her state for each of the five dimensions so that his or her state of health is described by five digits that take values from 1 to 5, with the state of health 11111 being considered a priori the best state of health and 55555 the worst state of health. The combination of these levels in each dimension defines a total of 3125 health states. The combination of the values of all the dimensions generates 5-digit numbers, with 243 possible combinations. It has proven to be valid in the Spanish population [49] and reliable in different populations [50] and the population of caregivers of other diseases 0.987 [51].Incremental cost-effectiveness ratio. For each group (experimental and control) the average cost of the intervention and the health effects will be calculated. The measurement of health effects is defined as Quality Adjusted Life Years (QALYs) [52]. QALYs are calculated by multiplying the years of life (life expectancy) by the quality of life of the participants.

#### 2.7.2. Focus Group Main Measure

Once the randomized controlled trial has been carried out and with its preliminary results, a focus group will be held, based on questions previously prepared by the research group, to determine the acceptability of the PI Integral-CARE. Likewise, key factors for its transfer to the public and private healthcare system will be identified. For this purpose, the different interventions made in the focus group will be recorded and transcribed. The transcripts will be analysed with the support of specific software for discourse analysis, which will be used to organize and classify the data into categories and then proceeds to sort and analyse them [53].

#### 2.7.3. Secondary Measures

(a)Socio-demographic data

Information on age, sex, income, time spent on caregiving, educational level, marital status, etc., will be collected.

Bem Sex Role Inventory (BSRI): according to the Bem Sex Role Inventory (BSRI) individuals can be classified as male, female, androgynous (both male and female), and undifferentiated (neither predominantly male nor predominantly female). This inventory consists of 60 adjectives of which 20 are stereotypically masculine, 20 are feminine, and another 20 have no gender typing. Each adjective is scored from 1 corresponding to never or rarely and 7 corresponding to always or almost always. Cronbach’s alpha coefficients were α = 0.86 for masculinity, α = 0.82 for femininity, for androgyny α = 0.85, and social desirability α = 0.75 [54].

Anthropometric measurements and body composition: anthropometric measurements will be taken under standardized conditions. Height (cm) and weight (kg) will be measured using a stadiometer (Seca 22, Hamburg, Germany). Waist circumference (cm) will be assessed at the midpoint between the ribs and the iliac crest, with the participant in a standing position (anthropometric tape, Harpenden Holtain, Crosswell, UK). Hip circumference (cm) will also be assessed. A bioimpedance meter (TANITA) will be used to assess body composition. Triceps skinfold (TFP): will be measured with a skinfold calliper (Lange^®^, Beta Technology, Santa Cruz, CA, USA) [55].

(b)Physical Area

The 2-min walking test: measures the maximum distance (in meters) that each participant can walk in 2 min along a rectangular route. It has high reliability, and the intraclass correlation coefficient is 0.888 [56].

Lower body strength: the 30 s Chair Stand Test will be performed, which consists of sitting down and standing up from a chair for 30 s. The test involves counting the number of times the participant can stand up completely from a seated position with the back straight and feet flat on the floor, without pushing off with the arms. The reliability coefficient of the 30 s Chair Stand Test is high (0.87) [57].

Upper body strength: will be evaluated through the Arm Curl Test, which consists of performing weighted arm flexion extensions. The test involves determining the number of times the participant can lift a weight by performing an arm flexion extension with a weight in the hand of 2.3 kg for women for 30 s. The Arm Curl Test obtained a reliability coefficient of 0.83 [57]. Additionally, a manual grip test will be performed using a digital dynamometer (TKK 5101 Grip-D; Takey, Tokyo, Japan) to assess manual grip strength [58]. Participants will perform a total of two attempts, alternately with both hands, of both tests. The best value of the two trials will be chosen for each hand and the average of both hands will be used for subsequent analysis.

Flexibility of the upper limb: will be evaluated through the Back Scratch Test which consists of reaching hands behind the back. This is a measure of the total shoulder range of motion and involves measuring the distance between the middle fingers or the overlap of the middle fingers behind the back using a ruler. The best score from a total of two attempts for each arm (in centimetres) will be recorded and the average of both arms will be used for further analysis [57]. The intraclass correlation coefficient is 0.925 [59].

Lower limb flexibility: will be measured by performing the Sit and Reach Test. Participants will be positioned in a seated position with one leg extended, and then slowly bend down by sliding their hands down the extended leg in an attempt to touch (or pass) the toes of the toe line. The number of centimetres before reaching (negative score) or beyond (positive score) the toe [60] will be recorded. Two trials will be measured with each leg and the best value for each leg will be recorded, and the average of both legs will be used in the analyses. The intraclass correlation coefficient value of the SRT is 0.92, showing high reliability [61].

Speed: the Brisk Walking Test will be used. This test consists of measuring the time that each participant takes to walk 30 m. Two repetitions will be performed with one minute of rest between them. The best result will be recorded [62], the test-retest reproducibility is 0.95 and the Cronbach’s alpha reproducibility coefficient is 0.96 [63].

Functional reach: the Functional Reach Test [64] will be used. The participant is placed next to a wall with the arms at 90 degrees to the trunk and will have to reach the maximum frontal distance and remain in that position for a few seconds, without altering its base of support. The maximum distance recorded perpendicular to the wall is recorded. The Cronbach’s alpha value is 0.81, so the reliability of the test is high.

Short Physical Performance Battery: Short Physical Performance Battery (SPPB) is a battery composed of 3 direct observation tests which are gait speed, balance, and time to get up 5 times from a chair [65]. The Cronbach’s alpha value is 0.70 [66].

Self-perceived physical fitness: the International Fitness Scale (IFIS) [67], which consists of five Likert scale questions on how participants perceive their general fitness, cardiorespiratory fitness, muscular strength, speed–agility and flexibility (“very poor”, “poor”, “average”, “good”, and “very good”) in comparison to their friends, will be used. The internal consistency of the IFIS scale obtained a value of 0.80 according to Cronbach’s alpha [68].

The Intensity and physical activity session control: the Xiaomi Mi Band 3 activity bracelet will monitor and record the heart rate as well as the activity performed during the session.

(c)Psychoemotional Area

Caregiver overload: will be evaluated through the Zarit Burden Inventory test, an instrument used to quantify the degree of overload suffered by caregivers of dependent adults. The Spanish version has 22 questions in Likert format. For each item, caregivers should indicate how often they feel using a scale of 1 (never), 2 (rarely), 3 (sometimes), 4 (quite often), and 5 (almost always). The scores obtained for each item will be summarized, and the final score will represent the degree of caregiver overload. Its internal consistency is high, with a Cronbach’s alpha coefficient of 0.91 [69].

Back pain: will be evaluated using the Roland-Morris questionnaire (RMQ) in its Spanish version. It is a simple and quick instrument composed of 24 questions that can be completed by the patient autonomously and which reflect limitations in different activities of daily living attributed by the participants to their back pain. They must mark each item that applies to their current condition. The scoring is also simple and quick; each marked item receives a score of 1, so the scores range from 0 (no disability caused by back pain) to 24 (maximum possible disability). The Cronbach’s alpha values were 0.8375 (day 1) and 0.9140 (day 15) in the validation [70].

Depression: will be assessed through the Geriatric Depression Scale (GDS) questionnaire. This questionnaire consists of 15 questions on how the participant has felt in the last 14 days and responses are limited to “yes” or “no”. The internal consistency analysis obtained an alpha coefficient of 0.87 [71].

Happiness and Life Satisfaction:

General Happiness Questionnaire [72]: is a global measure of subjective happiness, which evaluates a molar category of wellbeing as a global psychological phenomenon, considering the definition of happiness from the perspective of the respondent. It consists of 4 items with a Lickert-type response, its correction is made by adding the scores obtained and dividing them by the total number of items. The four items showed an internal consistency from good to excellent, demonstrating comparability between samples of different ages, occupations, languages, and cultures. The alphas ranged from 0.79 to 0.94 (Mean = 0.86).Satisfaction with Life Scale: the scale consists of 5 items, rated in the original scale from 1 to 7, and in the Spanish version from 1 to 5, but both scores range from “totally disagree” to “totally agree”. This version has good psychometric properties. The reliability index calculated for Cronbach’s alpha scale indicates that the scale has a very good internal consistency (α = 0.84) [73].

Satisfaction with Occupations and Occupational Balance: the aim is to characterize the person’s performance patterns and analyse his or her roles.

The Satisfaction with Daily Occupations questionnaire (SDO-13) is a tool that seeks to obtain information on the relevant aspects of occupation to understand the client’s satisfaction with the current performance of an occupation. The Cronbach’s alpha values of the SDO-13 scale are 0.80 and 0.88, demonstrating high internal consistency [74].The Occupational Balance Questionnaire (OBQ-E) consists of 13 statements to be answered using a Likert scale that is scored from 0 to 5, being 0, “completely disagree” to 5, “completely agree”, with the possibility to obtain a score from 0 to 65, and the higher the score, the higher the satisfaction with the occupations. The internal consistency obtained a value of Cronbach’s alpha = 0.948 [75].

Rosenberg self-esteem scale: evaluates the degree of satisfaction with oneself, estimating the distance between the ideal self and the real self of the individual. Its Spanish adaptation has a high internal consistency (Cronbach’s alpha = 0.87), high temporal reliability, and adequate validity [76].

Perceived social support: the Duke-UNC-11 Functional Social Support Questionnaire [77] will be used to determine the perceived social support of older caregivers. This scale has 11 items on a Likert response scale from 1 (“much less than I want”) to 5 (“as much as I want”). The score ranges from 11 to 55 points. In the Spanish version, a cut-off point was chosen at the 15th percentile, corresponding to a score of <32. A score equal to or higher than 32 indicates standard support, whereas less than 32 points indicates low perceived social support. In the Spanish population, the internal consistency was 0.90 [78].

Family Functionality: will be measured utilizing the Family Apgar scale, since it evaluates the perception of family functioning by exploring the respondent’s satisfaction with his/her family relationships. It consists of five Likert-type items (0 = almost never, 1 = sometimes, and 2 = almost always). The cut-off points are as follows: functional family: 7–10 points; mildly dysfunctional family: 4–6 points; and severely dysfunctional family: 0–3 points. The internal consistency measured by Cronbach’s alpha is high (0.84) [79].

(d)Health education and cognitive–behavioural area.

Cognitive–behavioural aspects: the Spanish version of the Cognitive–Behavioural Avoidance Scale (CBAS) will be used. It consists of 31 items reflecting different avoidance coping strategies, which load on four factors: behavioural/social, behavioural/non-social, cognitive/social, and cognitive/non-social. Response options are on a five-option Likert-type scale ranging from “Not so true for me” to “Extremely true for me”. A high score indicates more avoidance. Cronbach’s alpha of the CBAS was 0.89, demonstrating high internal consistency [79].

Alzheimer’s Disease Knowledge: the Alzheimer’s Disease Knowledge Scale (ADKS) will be administered, which contains 30 true/false items to assess knowledge about AD, and covers risk factors, assessment and diagnosis, symptoms, course, impact on life, care and treatment, and management [80].

(e)Adherence to the program: a record will be made of the number of sessions carried out.(f)State of health of the proxy patient:

Degree of dependence of the person with AD: this will be assessed using the Barthel Index, an instrument that measures a person’s ability to perform ten basic activities of daily living (ADLs), obtaining a quantitative estimate of his or her degree of independence. The values assigned to each activity are based on the time and amount of physical assistance required if the patient is unable to perform that activity [81]. As for the evaluation of internal consistency, it presents a Cronbach’s alpha of 0.86–0.92 [82].

Cognitive status of the patient: the patient’s cognitive status will be assessed using the Global Deterioration Scale (GDS). This is a scale consisting of a clinical description of seven distinct stages ranging from normal to the most severe degrees of AD. Its scoring is stage 1 (normal), stage 2 (subjective memory complaint), stage 3 (mild cognitive impairment), stage 4 (mild dementia), stage 5 (moderate dementia), stage 6 (moderately severe dementia), and stage 7 (severe dementia).

### 2.8. Statistical Analysis

Baseline characteristics of study participants will be presented as means (standard deviation) for continuous variables and proportions for categorical variables. Two types of analyses will be performed: (1) an intention-to-treat analysis, including all participants and (2) a per-protocol analysis, including only those participants who complete the entire study.

Intention-to-treat analysis: this analysis will include all randomized participants (in the groups to which they were randomly assigned) in the analysis. Multiple imputations will be used to impute missing data. The effects of the intervention on the primary and secondary variables will be assessed through repeated measures analyses of covariance adjusted for age and baseline values. The results will include the effect size (95% confidence interval) and statistical significance for each study measure concerning time and its interaction effects (group × time). Statistical significance will be set at the conventional level of *p* < 0.05. In the sensitivity analyses, the imputation of the data will be performed from the data of the patients at baseline and those who completed the study, to avoid estimation biases.Per protocol analysis: Analyses similar to those described above will be performed but only in those participants who attended at least 75% of the IP sessions.

### 2.9. Cost-Effectiveness Analysis

The Materials and Methods should be described with sufficient details to allow others to replicate and build on the published results. Please note that the publication of your manuscript implies that you must make all materials, data, computer code, and protocols associated with the publication available to readers. Please disclose at the submission stage any restrictions on the availability of materials or information. New methods and protocols should be described in detail while well-established methods can be briefly described and appropriately cited.

Research manuscripts reporting large datasets that are deposited in a publicly available database should specify where the data have been deposited and provide the relevant accession numbers. If the accession numbers have not yet been obtained at the time of submission, please state that they will be provided during review. They must be provided before publication. Interventional studies involving animals or humans, and other studies that require ethical approval, must list the authority that provided approval and the corresponding ethical approval code.

## 3. Discussion

To our knowledge, this would be the first research project to study the cost-effectiveness of an IP that would simultaneously include psycho-educational, cognitive–behavioural and physical training aspects to improve HRQoL in caregivers of people with AD. In addition, this project aims to address the sustainability of the social framework in which caregiving is carried out and to explore new avenues leading to a more equitable model that generates new opportunities for entrepreneurship and employment. The introduction of the concept of “caregiving” and social and community responsibility in the field of care will allow us to investigate new models of care for dependent persons. This will initiate new lines of action to promote social sustainability of the care system to avoid or delay the institutionalization of dependent persons.

Integral-CARE will be developed from an intergenerational perspective, through programmed interventions through Service-Learning. Participation focused on Service-Learning and intergenerational programs will encourage the development of new models of attention and care, as well as intergenerational coexistence and innovative initiatives of entrepreneurship and employment in this professional sector.

Moreover, if the effectiveness of this comprehensive programme in improving the quality of life of the caregivers of people with AD and its cost-effectiveness compared to conventional treatment (usual care) were demonstrated, it would represent an opportunity for economic savings for the health system. Likewise, the possibility of implementing Integral-CARE in different associations of relatives of people with AD would be studied, focusing on the applicability of the integral program in their centres. If it proves to be an effective program for improving the physical, psycho-emotional, cognitive–behavioural, and educational qualities of caregivers of people with AD, it could be a bet to promote social recognition and participation of caregivers of people with AD.

## 4. Conclusions

This project will investigate the applicability, safety, HRQoL, and incremental cost-effectiveness of a 9-month comprehensive interdisciplinary CARE program for caregivers of people with Alzheimer’s disease, as well as the transfer of the results obtained to the public and private healthcare economy in Extremadura. The results of this research will help to improve the efficiency of health services for caregivers of patients with Alzheimer’s disease. In addition, guidelines will be proposed for the transfer of the findings obtained to the Extremadura social and healthcare system and market. If the interventions prove to be effective, this study would be a viable bet to promote the social recognition and participation of AD caregivers.

## Figures and Tables

**Table 1 ijerph-19-15243-t001:** Session characterisation.

Intervention Area	Sessions (Number)	Modality	Duration (minutes)	Module
Physics	20	Face to face	50	Upper body strengthening
				Lower body strengthening
				Trunk strengthening
	10	Virtual	15	Flexibility
				Interesting aspects in this area
Psychoemotional	20	Face to face	50	Emotions
				Roles/Overhead
				Occupational balance
				Satisfaction/Happiness
	10	Virtual	15	Complementary alternatives for this area improvement.
				Interesting aspects in this area
				Knowledge of the disease
Cognitive–behavioural and health education	20	Face to face	50	Health education:Self-care and Social support
	10	Virtual	15	Interesting aspects in this area

**Table 2 ijerph-19-15243-t002:** Assessments scheduled for both experimental and control groups.

Assessment	Baseline	Month 3	Month 6	Month 9	Month 10
Applicability	X			X	X
Health-Related Quality of Life	X		X	X	X
Incremental cost-effectiveness ratio	X			X	X
Sociodemographic data	X				
Bem Sex Role Inventory (BSRI)	X	X	X	X	X
Anthropometric measurements and body composition	X	X	X	X	X
The 2-min walking test	X	X	X	X	X
Lower body strength	X	X	X	X	X
Upper body strength	X	X	X	X	X
Flexibility of the upper limb	X	X	X	X	X
Lower limb flexibility	X	X	X	X	X
Speed	X	X	X	X	X
Functional reach	X	X	X	X	X
Short Physical Performance Battery	X	X	X	X	X
Self-perceived physical fitness	X	X	X	X	X
Caregiver overload	X	X	X	X	X
Back pain	X	X	X	X	X
Depression	X	X	X	X	X
Happiness and Life Satisfaction	X	X	X	X	X
Satisfaction with Occupations and Occupational Balance	X	X	X	X	X
Rosenberg self-esteem scale	X	X	X	X	X
Perceived social support	X	X	X	X	X
Family Functionality	X	X	X	X	X
Cognitive–behavioural aspects	X	X	X	X	X
Alzheimer’s Disease Knowledge	X	X	X	X	X
Degree of dependence of the person with AD	X	X	X	X	X
Cognitive status of the person with AD	X	X	X	X	X

## Data Availability

Not applicable.

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
