# Peer review of "Cost-Effectiveness of the Comprehensive Interdisciplinary Program-Care in Informal Caregivers of People with Alzheimer’s Disease"

_ijerph, 2022, doi:10.3390/ijerph192215243_

Round 1

Reviewer 1 Report

1.  Introduction is too long and cost-effectiveness studies related to this paper are not reviewed well.

2. The context does not match the subtitle Cost-Effectiveness Analysis.

3. The spelling of HRQoL and HRQOL needs to be unified.

4. Based on CONSORT and SPIRIT checklist, data management, quality control, and proposed study timeline should be provided.

Author Response

Dear Reviewer,

We appreciate your comments and recommendations to improve our manuscript.

The detailed comments are as follows:

  1. Introduction is too long and cost-effectiveness studies related to this paper are not reviewed well.

We thank you for your appreciation and agree. The information provided in the introduction has been shortened. The bibliography used to develop the background has also been updated. In doing so, we believe that the introduction and contextualisation have improved the quality of the article.

  1. The context does not match the subtitle ‘Cost-Effectiveness Analysis’.

Thank you for your comment. We have redrafted the introduction and addressed in the contextualisation the importance of cost-benefit analysis using up-to-date literature.

  1. The spelling of ‘HRQoL’ and ‘HRQOL’ needs to be unified.

Thank you for this contribution, we have modified the acronyms in response to your comment.

  1. Based on CONSORT and SPIRIT checklist, data management, quality control, and proposed study timeline should be provided.

We thank you for your comment, and in this regard we have proceeded to include data management, quality control and the timeline of the study in the manuscript.

Reviewer 2 Report

Dear Authors, 

The study protocol manuscript titled "Cost-effectiveness of the comprehensive interdisciplinary Program-Care in informal caregivers of people with Alzheimer's disease" is well written and seems to be clearly organized. However, in my opinion, a better and updated references concerning the background of the study should be mentioned. Indeed only few papers of the last three years were cited (doi: 10.31887/DCNS.2009.11.2/hbrodaty; doi: 10.31887/DCNS.2000.2.2/mdavidson; doi: 10.14283/jpad.2022.53; doi: /10.1007/s40273-019-00788-3; doi: 10.1186/s13195-022-00969-x). On the other, since no study protocol like the one the Authors proposed are present in the main literature, this should be better underlined.

Moreover, I have some questions and suggestions on the materials and methods section in order to improved the quality of your paper. 

1- concerning the enrollment criteria, the Authors should underline that the study considers also age and gender as mentioned in 2.7.3 subheading, adding few sentences also in the 2.1 or 2.3 subheading. 

2- Did the Authors have preliminary results?

3- Please, add the full text for HRQOL (Health-Related Quality Of Life).

4- Please, always use the same abbreviations: "HRQOL" or "HRQoL".

5- Table 1: I suggest the Authors to divide each line, such as "Physics", "Phycoemotional" and "Cognitive-behavioral and health education", by a stripe or additional space in order to better distinguish the "Module" column.

Author Response

Dear Reviewer,

We appreciate your comments and recommendations to improve our manuscript.

The detailed comments are as follows:

REVIEWER 2

Dear Authors, 

The study protocol manuscript titled "Cost-effectiveness of the comprehensive interdisciplinary Program-Care in informal caregivers of people with Alzheimer's disease" is well written and seems to be clearly organized. However, in my opinion, a better and updated references concerning the background of the study should be mentioned. Indeed only few papers of the last three years were cited (doi: 10.31887/DCNS.2009.11.2/hbrodaty; doi: 10.31887/DCNS.2000.2.2/mdavidson; doi: 10.14283/jpad.2022.53; doi: /10.1007/s40273-019-00788-3; doi: 10.1186/s13195-022-00969-x). On the other, since no study protocol like the one the Authors proposed are present in the main literature, this should be better underlined.

Thank you for your appreciation. We have updated the bibliography used to develop the background to the study and we believe that this has enriched the introduction.

In addition, the discussion section has underlined the importance of being a pioneering study in the field of carers of people with Alzheimer's disease.

Moreover, I have some questions and suggestions on the materials and methods section in order to improved the quality of your paper. 

1- concerning the enrollment criteria, the Authors should underline that the study considers also age and gender as mentioned in 2.7.3 subheading, adding few sentences also in the 2.1 or 2.3 subheading. 

We welcome your comments. The measures of age and sex will form part of the socio-demographic data for the characterisation of the sample, however, they do not represent an admission criterion to form part of the sample. For this reason, both variables are considered secondary measures of the study.

2- Did the Authors have preliminary results?

The project is currently in the sample recruitment phase, so we do not have preliminary data from the comprehensive interdisciplinary care programme for informal carers of people with Alzheimer's disease.

3- Please, add the full text for HRQOL (Health-Related Quality Of Life).

Thank you for your appreciation, we have used "health-related quality of life" for the first time and indicated the acronym.

4- Please, always use the same abbreviations: "HRQOL" or "HRQoL".

Thank you for this contribution, we have modified the acronyms in response to your comment.

5- Table 1: I suggest the Authors to divide each line, such as "Physics", "Phycoemotional" and "Cognitive-behavioral and health education", by a stripe or additional space in order to better distinguish the "Module" column.

We agree with this, table 1 has been divided into modules.

Round 2

Reviewer 1 Report

Authors revised the manuscript well. I have no further comments.